# Methods Employed in Studies Identifying “Levels” of Test Anxiety in University Students: A Systematic Review

**DOI:** 10.3390/bs15030331

**Published:** 2025-03-07

**Authors:** Jerrell C. Cassady, Ser Hong Tan, Al Robiullah, Isabella Badzovski, Jessica Janiuk

**Affiliations:** 1Department of Educational Psychology, Teachers College, Ball State University, Muncie, IN 47306, USA; al.robiullah@bsu.edu (A.R.); isabella.badzovski@bsu.edu (I.B.); jessica.janiuk@bsu.edu (J.J.); 2Centre for Research in Child Development and Science of Learning in Education Centre, National Institute of Education, Nanyang Technological University, Singapore 637616, Singapore; serhong.tan@nie.edu.sg

**Keywords:** test anxiety, leveling strategies, systematic review, university students

## Abstract

Test anxiety research has been dominated by attention to theory building and examining the causes and consequences of this construct. However, recently, considerable attention has been turned toward using test anxiety as a diagnostic indicator of students who are at risk of underperforming in educational settings. This systematic review of the literature was focused on (a) describing the strategies used in the field, (b) highlighting the benefits and limitations of those approaches, and (c) offering guidance in creating a framework for appropriate methods when identifying severity levels on measures of test anxiety for university students. The results confirmed that the vast majority of studies on test anxiety have no formal “leveling” approaches (maintaining test anxiety as a continuous variable). However, when researchers do employ “leveling” strategies, the majority adopt inappropriate methods (e.g., single-sample splits). However, there are exemplars that demonstrate statistically sound procedures for identifying distinct profiles of test anxiety that may form a basis to build consensus around a classification method for elevated test anxiety.

## 1. Introduction

Test anxiety has been an area of vibrant research for over 50 years but has experienced a considerable increase in interest as assessments have become more prevalent in educational decisions regarding student progress, instructor efficacy, and institutional success ([33]; [60]). The research on test anxiety and learner experience has demonstrated compelling and recurring evidence that there are negative relationships among test anxiety and student performance ([47]), the use of effective study strategies ([1]; [59]), and confidence or self-efficacy over academic evaluations ([10]). Furthermore, test anxiety has been positively linked with negative affective variables including preclinical levels of depression, anxiety, and neuroticism ([15]; [68]; [69]), ratings of general well-being ([62]), and maladaptive coping strategies ([7]; [28]; [47]; [53]). While the general findings in the field support an overall negative relationship between test anxiety and academic success ([18]), there is growing evidence that this representation may be oversimplified. That is, non-linear relationships tend to a provide superior model fit ([13]), suggesting that low-to-moderate test anxiety may have a “facilitative” influence ([27]; [50]).

The evidence of incidence rates for test anxiety has demonstrated that the reported rates of occurrence have been on the rise over the past 20 years. Reports in the early 2000s typically suggested that between 15 and 30% of students in university settings were likely to report experiencing test anxiety ([6]; [35]; [46]). However, recent analyses have identified that the reported rates of test anxiety have increased significantly in the past 10 years—with as many as 80% of university students reporting they experience moderate-to-severe levels of test anxiety ([12]). The data also indicate that the reported level of test anxiety tends to increase across the educational timeline, with a higher proportion of university students reporting heightened test anxiety than students in elementary and secondary education settings ([45]).

The dominant method for evaluating and assessing test anxiety has been self-report measures since Sarason and Sarason offered the Test Anxiety Questionnaire in 1952. Over that time frame, self-report instruments have almost universally relied upon using total scores or subscale values of componential aspects of test anxiety that meet the assumptions of continuity and normality as well as align with a theoretical model of test anxiety ([31]; [48]; [51]; [54]). This approach to building the field of test anxiety research has bolstered several viable theoretical representations for the test anxiety construct, typically focusing on identifying the causes and consequences of test anxiety in traditional educational settings ([61]; [69]). A less vibrant domain of research in the field (particularly for post-secondary students) has been focused on using these instruments to document the efficacy of interventions designed to relieve or mitigate the experience of test anxiety over time ([21]; [24]; [53]). In most of these intervention studies, there continues to be a primary focus on identifying success by measuring the “change” in measured levels of test anxiety on that continuous representation.

While theory development has been strong over the past half century, the recognition of increased rates of incidence and levels of severity in reported test anxiety among learners has created a groundswell of interest among education professionals, students, and parents to devote more attention to a practical orientation focusing on identification, intervention, and treatment efforts for learners who need support to reach optimal performance (see [24]; [53]). This work is driven by awareness that interventions to support these learners in educational settings can increase student retention rates and promote greater degrees of student wellness and success ([60]; [44]). This work has been more prevalent in elementary through secondary school settings, perhaps driven by the greater focus of school support personnel to mitigate the social and emotional needs of learners. However, there is a growing interest in developing similar identification and intervention supports for learners in higher education settings, prompting our interest in reviewing the field to help form a consensus around effective methods to determine what students are “at risk” for academic difficulty due to test anxiety.

The purpose of this systematic review was to investigate studies in the literature that have employed categorical strategies for identifying “levels” of test anxiety in post-secondary populations. Using this corpus of the existing literature as a baseline, our goals are to identify existing trends and offer recommendations for best practices. Our choice of a systematic review, as opposed to other review methodologies (i.e., scoping review), was based on the primary purpose of the study to synthesize existing research in this population ([37]). Before identifying the trends in the field, we discuss the primary barriers to generating durable and valid categorical representations of test anxiety.

### 1.1. The Structure of Test Anxiety

Research in the field of test anxiety traditionally supported a two-factor representation, where “overall” test anxiety comprised two subordinate and related dimensional constructs. The classic terms for these subcomponents of test anxiety were Worry and Emotionality ([31]), which have more recently also been discussed as “cognitive” and “physiological” dimensions ([9]). Recent approaches to measuring test anxiety have also incorporated a “social” dimension of test anxiety ([23]; [34]) or made more fine-grained distinctions among aspects of test anxiety (e.g., Tension, Worry, Cognitive Interference, and Physiological Indicators; [48]). Motivational constructs underlying the stressors faced in academic settings have also been proposed as critical determinants in the development or manifestation of test anxiety (e.g., Control–Value Theory; see [41]; [58]). These broader representations of multiple dimensions continue to support a more comprehensive view of contexts, histories, and beliefs of learners. However, there continues to be debate regarding which dimensions are distinct “types” of test anxiety and which are co-occurring psychological or ability constructs (see [43], for further discussion).

Research has routinely identified that the various dimensions of test anxiety are positively and significantly related to one another ([9]; [24]; [23]; [34]). However, the research also demonstrates differential influences of those factors on outcome variables such as student performance. [70]’s ([70]) “additive model” indicated that the cognitive and physiological dimensions exerted unique and negative impacts on performance. Subsequent research specified that the cognitive domain (e.g., Worry and Cognitive Interference) tends to be more directly tied to student performance declines, presumably due to the attention to barriers to effectively or efficiently encoding, storing, or retrieving the primary content either in the study or testing context ([14]; [9]; [22]; [43]). Another interpretation of the “combined” influence of cognitive and physiological indicators of test anxiety suggested that the physiological dimension serves as an “alert” mechanism, identifying a perceived threat in the academic setting, which subsequently activates the cognitive dimension ([24]). In both of these representations, there is a tendency to focus on the cognitive aspects in a more “trait-like” representation (relatively stable over time and context) and the physiological aspect as more “state-like” (varied based on the task at hand; [9]). Ultimately, most practitioners using assessment tools based on bifactor or multidimensional models of test anxiety tend to focus on “composite” or “total” scores to identify individuals with greater need; at the very least, this ensures that variations in manifestation are equitably included in the determination of test anxiety level ([23]; [33]; [51]; [62]).

Naturally, variations in representation of the test anxiety construct as well as differences across measures in the discipline provide a clear challenge to establishing a universal agreement in determining clear “levels of test anxiety.” This is further exacerbated by the reality that there is no clinical definition for “test anxiety” accepted by professional psychologists and psychiatrists (e.g., [2]; [66]). That is, the defining characteristics of test anxiety—as defined by theoretical models of test anxiety—are not clearly translated to defining a clinical disorder. A diagnosis of “clinical test anxiety” is typically made by clinicians through a combination of screening instruments and clinical interviews ([25]). Furthermore, there is no established criterion for how prevalent test anxiety “should be” in the general population or what level of reported anxiety dictates elevated levels across measures ([30]; [65]). The absence of a “clinical sample” in studies has precluded the ability to perform a strong test of different models’ or measures’ precision in detecting truly detrimental levels of test anxiety ([25]).

As such, researchers and practitioners who employ different measures of test anxiety are essentially examining different representations of the construct, creating a challenge for making reasonable comparisons across studies regarding the “levels” of test anxiety. A similarly troubling trend in the research is where measures that are not specifically designed to measure test anxiety are used to make conclusions about test anxiety. For instance, several studies over the years ([5]) have used the State-Trait Anxiety Inventory (STAI; [55]), to identify “levels of test anxiety” in their sample. Comparing those results to studies that employ another measure (e.g., the Test Anxiety Inventory, TAI; [54]) will likely generate incompatible representations of the populations, because the STAI is a “general anxiety” scale focused on trait and state subcomponents (not focused on testing), while the TAI focuses on the bifactor model of test anxiety (i.e., Worry and Emotionality over examinations specifically).

Our systematic review cannot hope to reconcile the variations of assessment instruments, and there is no intent to suggest that one assessment tool or theoretical model for test anxiety is preferred over others. The point of this review is to call attention to the factors that influence the variations in outcomes observed in the literature to guide researchers and practitioners alike to identify reasonable conclusions on identifying levels of severity for learners with test anxiety.

### 1.2. Measurement Challenges for Test Anxiety Scales

In addition to challenges imposed by varied assessment models for test anxiety and the imprecise definitions for “clinical levels” of test anxiety, traditional measurement issues also challenge this field of research. A fundamental assumption of most test anxiety scales is that the measure generates a total scale value across which incremental levels of test anxiety are measured in a relative interval fashion. This presumption does not necessarily align with classic assessment theory because most surveys use Likert-type items, which may not meet assumptions of interval measurement ([19]). When directly tested, the scales generally demonstrate reasonable normality and consistency with expectations of interval measurement. The reality of the assessments is that they are based on a collection of responses to essentially ordinal data. Furthermore, studies examining differential item functioning (DIF) for test anxiety scales often report that not all items contribute equitably to the latent test anxiety construct ([3]; [62]), calling into question simple reliance on “total score” tabulation for determining a critical cut point. That is, validity threats are increased because a total scale score may misrepresent the latent test anxiety factor by undervaluing or overvaluing certain items.

A final historical barrier to the assessment of test anxiety has been the difficulties that often arise when instruments are composed of reverse-coded items or when inverting the valence of specific items in a scale. That is, some scales have items that focus on “high indication of anxiety” as well as those that are proposed to identify “low anxiety” with the direction to reverse-code and compile the values into a single factor ([51]; [9]). Work examining these approaches demonstrate that a reasonable factor solution may be derived, but the response patterns generally indicate that the reverse-coded items actually generate a secondary factor more representative of “test confidence” and should not be merely recoded and added to the total scale score (see [9]; [20]).

### 1.3. Leveling Strategies

The overwhelming approach to research focuses on creating “level” attempts to identify “high” and “low” test anxiety groups within a population. The appeal of this approach beyond identifying learners with needs for support services is a simplified method of evaluating group differences (e.g., performance differences for high- and low-test-anxiety groups). Strategies employed for identifying groups of individuals based on the level or degree of a psychological construct typically fall into two broad methodological strategies. The first strategy (cf., “Expert-Based”) is dependent on experts indicating (a) the level of a construct needed to meet a “heightened” or “clinical” threshold and (b) how that threshold can be estimated by a given measure ([16]). As mentioned before, this is particularly difficult with test anxiety due to the absence of a clinical definition for the construct ([25]). The second approach (“Data-Based”) employs data-driven methods that establish group membership based on response trends in the target population ([52]).

#### 1.3.1. Expert-Based Methods

*Expert Ratings.* The expert-based methods operate under the assumption that an expert in the field can clearly identify that an individual exhibits symptoms consistent with a “clinical level” of that construct ([16]). One common strategy to accomplish this involves the Delphi Method, which involves experts indicating which items on a scale are most critical in identifying key diagnostic criteria as well as determining response levels that should be deemed clinically significant ([36]). When the construct of interest has a clear clinical definition (e.g., Depression or Generalized Anxiety Disorder), this process is validated by comparing scores for a confirmed clinical population against a non-clinical population to demonstrate that the scale effectively differentiates ([16]). However, with no clinical definition for test anxiety, this final validation step is impossible ([52]).

*Pre-determined cut scores.* Another strategy driven by expert judgments involves selecting a priori performance or rating levels on the measurement scale in question to determine levels of test anxiety. In a strategy similar to the Delphi Method, these expert-based judgments could be driven by clear theoretical alignment. However, the evidence is clear that pre-determined cut scores are often determined by arbitrary decisions based on scaling. An example of this approach to pre-determined cut scores is [4]’s ([4]) determination of levels of test anxiety using the Westside Test Anxiety Scale. Specifically, using the 5-point rating scale, students were deemed to represent five categories of test anxiety based on their average response to items falling into whole number ranges: 1.0–1.9 = Comfortably Low; 2.0–2.5 = Normal/Average; 2.5–2.9: High/Normal; 3.0–3.4 = Moderately High; 3.5–3.9 = High; and 4.0–5.0 = Extremely High test anxiety. These logical groupings are common across the field when a priori decisions are made, but without any empirical evidence that there is a clear differentiation among the identified levels, these findings are often unsupported or not validated.

#### 1.3.2. Data-Based Methods

Given the absence of an agreed-upon definition for “clinical” test anxiety, there is a significant preference for data-based strategies when establishing leveling criteria for test anxiety assessment tools. Data-based methods use patterns in respondent data to dictate group membership, varying widely in the levels of statistical sophistication and theoretical basis across studies and over the years. We recognize four general approaches that have dominated data-based approaches, including large-scale norming samples, “local sample” splits, item-focused group differentiation strategies, and person-centered group differentiation approaches.

*Large-scale norming samples.* The classic data-based approach to standard settings with assessment instruments involves gathering a large, representative sample with the target instrument and identifying the distribution of responses for the normal population. However, as many different areas of measurement theory have demonstrated, there are considerable barriers in this “basic” approach advocated in most classic test theory designs. First, developmental, cultural, racial, and gender variations in responses to individual items or in the distribution of total scores on assessment tools provide important reservations when performing these comparative analyses ([26]). Given that there are commonly reported differences in test anxiety averages reported across age, gender, first-generation university student status, and race ([29]; [46]; [45]; [61]), separate norming tables would be required. Given these barriers, it is not surprising that few attempts have been made in this way. A notable exception to this trend for test anxiety measures exists with the Test Anxiety Inventory ([54]), but the norms established for that measure are over 40 years old, and comparative analyses have confirmed variations in the contemporary student performance on that scale ([56]).

*Local sample splitting.* A far more common approach to using a data-focused approach to form test anxiety groups involves using the sample in a single study as the source data. This approach has been particularly common in intervention studies that attempt to examine the impact of a treatment approach to relieve test anxiety and explore the differential utility of that intervention based on the level of test anxiety or use the test anxiety measure to identify those students who will receive the intervention.

These strategies may involve using percentile splits (e.g., median, tertile, or quartile) to differentiate among learners or the sample mean and standard deviation to generate a performance-based criterion (e.g., one standard deviation over the mean suggests “High Test Anxiety”) ([17]; [38]). The obvious limitation to this broad approach is that the “cut scores” for each level of test anxiety will vary across studies, limiting comparability and generalization. This approach is also subject to flawed assumptions about the nature of test anxiety regarding the distribution of test anxiety in the population, the distinctiveness of the identified categories from one another, and the conceptualization of test anxiety itself (the measures chosen for a specific study will dictate the model of test anxiety assumed, which may vary greatly across studies).

*Scale- and item-focused group differentiation.* Two data-centric approaches that are primarily oriented toward response patterns on the given measurement tools for establishing profiles of test anxiety are Receiver Operating Characteristic curves (ROC) and the Rasch Rating Scale Method (RSM). These methods are both theoretically possible with test anxiety studies but are not commonly employed. The ROC method is consistent with traditional signal detection theory strategies that identify the rates of accurately identifying the presence of “the diagnosed condition” with an assessment instrument (taking into consideration rates of true and false positives and negatives) ([39]). This method can be used to adjust the cut point on a measure that maximizes the number of “accurate” identifications. However, the notable limitation with using ROC for test anxiety is the requirement of a clinically identified subpopulation upon which those cut score decisions are validated. This is generally achieved in studies using ROC by using an expert judge to determine the “clinical group” (see [63], for example).

The limiting factor imposed by the ROC’s need for a “clinical group” is overcome by the RSM, which also has the distinction of being a model focused on individual item characteristics rather than total scale performance levels. In a comparative study examining the accuracy of identifying individuals with ROC, RSM, and traditional standardized scores (e.g., z-scores), [19] ([19]) demonstrated that the results of RSM were as good or better than ROC and standardized test approaches, with the advantages of no required clinical sample and ability to estimate a “clinical” threshold for each individual student without being based on the sample population.

*Person-Centered Group Identification.* Ready access to statistical applications that support advanced modeling approaches has led to a sharp increase in the use of latent class analyses and cluster methodologies to identify learners with differentiable levels of expressed test anxiety ([42]; [59]). These strategies identify overall learner response patterns on one or more measures of test anxiety, identifying subgroups within the tested population who differ from one another in reliable ways. These analyses are largely isolated from preconceived biases in the expected number or composition of groups of test anxiety given the data-centric nature of the analyses.

## 2. Materials and Methods

The current study was designed as a systematic literature review exploring published peer-review research reporting analyses that either generated or relied upon subgroups of students based on their “level” of test anxiety. The focus of this investigation was limited to post-secondary students (i.e., university, college, and community college). The initial capture of articles for review occurred in April of 2022 and focused on a 10-year search (capturing all publications indexed with a publication year of 2012 or later). Delays in completing the work led to a re-examination of available articles in June 2024, augmenting the list to capture the period of time from 2012 to June 2024.

The search for relevant articles was conducted with support from a university librarian with expertise in framing large-scale record search and acquisition studies. Given the primacy of education and psychology in the research on test anxiety and the variations of language used to explain or discuss the construct, we employed multiple databases and search strings. The search databases used in this study were Academic Search Complete, APA PsycINFO, eBook Collection (EBSCOhost), ERIC, Web of Science, and the Psychology and Behavioral Sciences Collection (all searched in parallel). For search terms, to limit the overlap with other forms of anxiety, we were explicit in attempting to orient test, exam, or evaluation anxiety as the central construct but recognized the need to review both “anxiety” and “stress” as keywords that are common to the literature. The search terms used in all the databases included the following parameters, with a Boolean search string:Test AND (anxiety OR stress);Examination AND (anxiety OR stress);Evaluation AND (anxiety OR stress);For all searches, a limiter on the database for “university OR college” as the sample population was applied when enabled.

Once the searches were completed for all the databases, 1707 articles were identified for review (see Figure 1; [40]). Initial screening of this list was conducted to eliminate duplicates captured in different databases due to variations in naming or citation. This cleaning of the datafile resulted in 1669 unique articles available for screening. The second stage of screening involved accessing full-text versions of the referenced studies and verifying that the study in question measured test anxiety (e.g., excluding review articles discussing test anxiety). This reduced the number of articles for review to 435 (1129 of the list did not directly measure test anxiety, and 105 were inaccessible in full-text format or only available in a language our team could not reliably translate).

The next step in the review process engaged in examining if the study in question used a “leveling” procedure. Of those 435 full-text studies reviewed, 373 did not employ a leveling procedure to the test anxiety measure. These studies primarily employed the measure of test anxiety as a continuous dependent variable (e.g., used in descriptive, correlational, or causal–comparative analyses) or a continuous predictor variable in a regression design. While these studies did not explicitly determine a standard to be reached as evidence of “high test anxiety”, the articles often used high values on a continuous measure being predictive of some outcome as impetus to suggest conclusions such as “students with high test anxiety were more likely to….”. While these studies were not the primary interest in this investigation, we classified them as “continuous” measure studies to capture this very common trend in the field and call attention to the trend in the field, mostly to alert readers to the tendency in the field to use the term “high test anxiety” without having a specific definition to support that terminology. To be sure, this should not be seen as a critique of that work. Indeed, research on the construct of test anxiety and how it interacts with other psychological or educational variables is often most effectively conducted when using a continuous measure of test anxiety. Among the myriad benefits of that approach is the ability to detect curvilinear relationships among test anxiety and related variables. However, given our focus on examining strategies employed to identify students with categorically differing levels of test anxiety, these studies lie outside our domain of inquiry.

Consequently, the review of articles resulted in 62 studies that measured test anxiety and documented different levels of test anxiety for the participants. These 62 studies are the primary focus in this investigation, but it is instructive to reflect upon the finding that with the large body of available published research on test anxiety in post-secondary learners over the past 12 years (n = 1669), only 26.1% (n = 435) were verifiable as empirical studies measuring test anxiety, and only 3.7% (n = 62) explicitly identified and explored test anxiety levels (see Figure 1). In short, there is a vibrant discussion about test anxiety in the literature, but there is relatively little research that directly examines the “level” of test anxiety.

Once there was agreement in the coding team that articles met the requirements of our targeted inquiry focused on using leveling procedures to identify groups of test-anxious students, each study was once again reviewed to identify the methodologies employed for leveling as well as the measures employed most commonly. Two independent coders examined each article and provided their interpretation of the reported method. The following list of five distinct categories was created among the team after reviewing the studies and was used in the independent ratings. It is important to note that we had a 6th category identified (item- or scale-based grouping) that was sought (i.e., RSM and ROC methods), but no studies using a university sample were identified to use that methodology.

Continuous scale or no leveling: Articles that discuss test anxiety without specifying high or low levels of anxiety but may refer to trends related to high or low anxiety without specific criteria (e.g., regression analyses and path model studies). This was not a “leveling” strategy but was scored to identify the magnitude of this trend in the field.Local sample splits: Studies where group membership in the specific study was based on performance relative to that sample only (e.g., median, tertile, quartile splits, and mean ± standard deviation). These local sample splits typically created subgroups of similar sizes (as mandated by the reliance on only the specific study’s sample to determine groups).Published or validated splits: Articles that relied on pre-established and validated criteria to categorize participants’ test anxiety levels.Person-focused data-based grouping (e.g., “Cluster”): This approach relied on cluster methods, latent class analyses, profile analyses, and related strategies that employ student responses to classify people.Scale-based cut scores (“Logical Cut”): This approach establishes levels of test anxiety using a logical cut score based on scale values, typically without relying on the specific data from the study participants. Expert judgments often play a role in defining these scores

Cohen’s Kappa calculations were examined to ensure ratings were consistent (see Section 3 for Kappa values). Reconciliation for the few discrepancies in coding was achieved through discussion, with full agreement achieved.

## 3. Results

### 3.1. Leveling Strategies

Four distinct leveling strategies were identified in the studies captured in this review. Compared with related work in leveling research using psychological assessment instruments, these four strategies capture all the methods outlined except for (a) the expert-based method relying on comparison with clinical samples or the method of Delphi and (b) the data-based method using scale-focused group determinations (e.g., ROC and RSM). Naturally, the clinical comparison approach is limited by the absence of a clear clinical definition of test anxiety. As for the use of ROC and RSM, we are aware of studies using these strategies with younger populations ([63]) as well as with measures that are similar to—but not specifically—test anxiety ([19]). However, when examining the classification of university students into groups based on their level of test anxiety, we found no examples of that approach being employed in the published literature, suggesting a viable domain of future research, perhaps by comparing modeling methods to see if there is convergence in group identification.

Below, the identified number of studies (and overall percentage) are provided with an explanation of the general strategies employed. For more details on specific studies and their identified classification level, please refer to Table A1. It is instructive to clarify that the studies below have been classified by the instrument that was used in the study to “create” the different levels of anxiety. In some cases, multiple measures of test anxiety were collected, but the attention in the sorting task in this study was focused on the method used to establish a leveling decision. Those studies that employed multiple scales generally used the secondary scale as a validation check (i.e., levels established with Measure A generated group means that were significantly different on Measure B).

*Scale-based cut scores (“Logical Cut”).* The most commonly identified strategy used to identify levels of test anxiety in this review used a “scale-based cut score” approach based on an arbitrary criterion process that did not employ data-driven procedures to establish levels of test anxiety (n = 31, 50% of leveling studies). These cut scores were generally established based on the scale values for a measure, using a logical criterion that was not informed or influenced by the responses of the participants in the study in question. These may be characterized in many cases as involving or including expert judgments as well, as the researchers engaged in the process of determining the cut scores were clearly experts in either data analyses or test anxiety as a construct. As mentioned before, [4] ([4]) employed this method with the Westside Test Anxiety Scale. Another example of this approach was carried out by the authors [67] ([67]), who reported using student responses to 25 true–false items based on Sarason’s anxiety measure (score range = 0–25). They offered that values of less than 12 indicated minor anxiety, 12–20 demonstrated moderate anxiety, and greater than 20 counted as severe.

Evaluation of the inter-rater reliability for identifying studies as fitting into this category employed Cohen’s Kappa, which resulted in a value of 0.76 (indicating a strong level of agreement). A review of misaligned classifications indicated that those errors were generally due to classifying these studies as using sample-based splitting strategies (which will be described next).

*Local sample splits (“Splits”).* Another common approach was to employ sample splitting techniques using only the participants for the study in question (n = 10, 16.1% of leveled studies). These strategies include creating median, tertile, or quartile splits (generating equivalent subgroups) or using the mean and standard deviation of the study sample to identify extreme groups. In contrast to the scale-based cut scores, the sample splitting strategies are generally used with no attention to the potential range of responses. Cohen’s Kappa for this group of studies was 0.68, with initial disagreement primarily occurring with the scale-based cut scores.

*Person-focused data-based grouping (“Cluster”).* A qualitatively different method of using a study sample to determine different levels of test anxiety relies on clustering, latent class, latent profile, or other statistically derived group classification strategies (n = 10, 16.1% of leveled studies). These approaches differ from the sample splitting methods in that the groups identified through clustering, latent class, or profile analyses focus primarily on identifying data-derived differences among the respondents in a sample. As such, group membership is independent of the potential range of the scale, the number of groups is determined empirically, and the distribution of the sample among the groups is not pre-determined with respect to size or percentage of the population. Our team of independent coders had a Cohen’s Kappa agreement value of 1.0, indicating full agreement in initially classifying the studies using latent classification strategies.

*Standardized or published criteria (“Validated”).* The final defined method of establishing groups based on levels of test anxiety identified in the sample was the use of pre-determined severity standards that were previously established through empirical strategies (n = 11, 17.8% of leveled studies). That is, these studies made use of a previously validated data-driven strategy for identifying learners at various levels of test anxiety. This approach could be characterized as having the highest level of external validity (among those identified in our review) because the grouping is based on a sample other than the one in the current study. This strategy also does not have a predetermined or sample-based limitation in the number of respondents who will be classified at various levels of test anxiety.

However, it is critical to recognize that the previously determined severity standards referenced in these 11 studies are also subject to limitations. That is, if the source criteria used were based on limited samples or inappropriate methodological approaches, a ripple effect in misclassification could occur. Cohen’s Kappa for rater agreement in identifying studies using this approach was 0.88, demonstrating high agreement.

### 3.2. Use of Test Anxiety Measures

To help summarize the overall representation of the field, our team also examined the specific measures used in the field to explore leveling strategies. As demonstrated in Table 1, the Test Anxiety Inventory was the most commonly used scale in the last 12 years when researchers were seeking to identify different “levels” of test anxiety. While the TAI has validated existing identified levels ([54]), not all studies in our review used those criteria (as shown, both logical cut and validated strategies were used). There is no clear evidence that the standards established in 1980 are still relevant to the current population (see [56]). Collectively, a review of Table 1 illustrates the diversity of measures as well as methods for determining the severity levels of test anxiety reported by participants in post-secondary settings. A more specific review of the 62 studies that provided specific leveling data is provided in Table A1, which can be used to identify the specific studies that employed the various methods of leveling for the interested researcher or practitioner.

## 4. Discussion

This systematic review of studies employing “leveling strategies” in university samples was undertaken to provide a documentation of the strategies that have been used in the field to date as well as provide recommendations to support more valid and consistent methods for identifying and subsequently supporting individuals in post-secondary education settings. Prior research focused on the influence of test anxiety has repeatedly demonstrated that university students report high levels of test anxiety ([9]) and that there are durable and reliable negative relationships between test anxiety and students’ study habits, coping strategies, and academic performance when levels of anxiety are elevated. This body of research has demonstrated reliable predictive power for identifying learners who are more likely to underperform as well as withdraw from university settings. Our focus in this work is to provide a pathway to identification and intervention such that early indicators of high test anxiety or poor academic readiness indicators can be detected early in the university experience to connect learners with interventions that will promote student retention and success ([11]).

### 4.1. Limited Attention Identifying “Elevated” Levels

The results of our review of the literature initially demonstrated a considerably low amount of explicit attention to true “leveling” in the field. That is, only 62 of the 435 studies that explicitly measured test anxiety in learners in post-secondary settings (14.3%) in the past 12 years were identified in our review of the literature to use direct leveling strategies to identify different proposed “groups” of students based on their level of test anxiety. The remaining 373 studies largely employ test anxiety as a continuous variable, typically examining the level of test anxiety in a regression model or as the outcome variable of interest. As the first point of interest, we raise this as a possible domain of deeper inquiry in the field. There is considerably more attention to identification and intervention efforts in students from Grades 1–12 (see [57]), despite evidence that the severity and prevalence of test anxiety tends to increase as students progress into university ([9]; [49]; [64]). We anticipate that this gap in the focus on identifying and serving students will rapidly decrease as universities continue to find methods to better serve and subsequently retain more students who arrive with less developed study habits and strategies that are known to boost success in post-secondary settings. While there is great theoretical benefit to learning how test anxiety influences learners’ outcomes in educational settings by exploring test anxiety as a continuous measure, for the simplicity of diagnosis often sought in programs designed to identify and serve “at risk” students, continued efforts to building a base of empirically driven strategies for classifying learners with differing levels of test anxiety shows great promise. However, as outlined next, we assert that careful attention to the methods of forming these classification groups is critical to building a consensus on what constitutes a clinical need in test anxiety.

### 4.2. Methodological Recommendations for Identifying “Levels” of Test Anxiety

From a methodological perspective, our primary finding is that the bulk of work that makes use of a leveling strategy for test anxiety has been employing questionable strategies. That is, the overwhelming majority of studies employed two strategies that we propose are not preferred for building a disciplinary approach to identify a meaningful criterion for defining “elevated” or “high” test anxiety among post-secondary students. The two most common strategies employed arbitrary cut scores and “local splits” to determine levels. The measurement concerns with both of these strategies are clear and have been established in the field for years ([16]; [52]). Put simply, the result for studies that use local splits is merely an identification of the “most test anxious people in the current sample”, which will vary from study to study. For the studies that employ an arbitrary cut score (usually using the average value for each item, e.g., [4]), the risks are based on the potential that each item does not have equitable value in determining the underlying construct or that the cut points are not realistically aligned with any indication of severity. For the latter strategy, the absence of an explicit definition or clinical criterion to validate the estimated cut scores further hampers the ability to establish meaningful validity.

This overarching limitation in the vast majority of studies poses a risk to having an established criterion for determining levels of severity for test anxiety and subsequently calls into question any reported estimates of the incidence of test anxiety in the university population. That is, while researchers have identified projections of the percentage of students who report moderate to high levels of test anxiety ([12]), these estimations are only relevant within the context of the measure in use and have to be reviewed within the context of the measures employed as well as the methods of labeling levels of test anxiety.

However, rather than simply projecting a negative narrative about using test anxiety measures to identify levels of need in the post-secondary student population, we see a key opportunity to build consensus in defining “high test anxiety” as a criterion-based characteristic that clearly identifies individuals who have a measured level of test anxiety that puts them at risk for learning loss in the university setting. The evidence from the few studies that have employed person-centered strategies (e.g., cluster methodology and latent-class analyses) demonstrate that there are measurable differences among learners in higher education that can be used to identify clear cut scores that indicate qualitatively different perceptions of threat or stress in academic assessments ([32]; [59]). To that end, we propose that research with a focus on identifying learners with levels of test anxiety that are likely to lead to academic challenges (either through performance or inappropriate study strategies and coping behaviors) should employ a few basic minimum strategies.

*Recommendation 1: Reduce the Use of Local Splits.* Our first recommendation is to eliminate the simplified method of using a local sample to set the “standards” for determining those with and without elevated test anxiety. Aside from the inherent changes that arise whenever you use a local and changing sample to identify “high test anxiety”, using local splits reduces the ability to compare the efficacy of interventions across settings. Intervention methods that prove highly useful in one setting may be less effective when the strategy for identifying the recipients of that intervention approach differs. Furthermore, the data have repeatedly demonstrated that there is usually a disproportionate number of learners who fall into the different categories of test anxiety (see [9]; [47]), but local sample splits almost invariably assign even numbers of students to their different “levels”. The only exceptions to this recommendation arise when programs are in the initial phases of attempting to identify their high-need population with a new assessment process or when resources are dramatically limited (and only a very small proportion of students can be supported).

*Recommendation 2: Eliminate Arbitrary Cuts.* Our recommendation about arbitrary cut scores is more extreme based on our experience reviewing the field. A prototypical arbitrary cut process involves identifying the range of potential responses and then dividing the level of severity based on the original response scale options. For instance, if a 4-point scale is used, the “high anxiety” group would often be identified by anyone who averages 3.5 on that 4-point scale overall. Setting logically based cut scores simultaneously makes weak items overly important and strong items undervalued in the identification of the psychological construct ([16]; [52]). These logical splits were traditionally valuable, but in the current era of ready access to more effective statistical solutions to test the relative utility of each item in determining the total scale, these approaches are simply outdated.

*Recommendation 3: Adopt and Adapt.* Overall, our suggestion for carrying out this leveling work in the field is to build upon existing and validated approaches with existing scales (with the explanation that this does not mean replicating cut scores identified in the prior two recommendations). When determining the viability of adopting a prior finding, we recommend that the method used in the original study should have been based on a representative or generalizable population (relevant to the population in question) and that the conditions of establishing the criterion scores can be replicated and validated. With the studies reported in this systematic review, the most promise was shown for the cluster and latent class (person-centered strategies). With sufficient sampling, these strategies allow a greater ability to clearly differentiate among learners to provide categorical identifiers for levels of test anxiety. An extension of this strategy that has not been identified in the current literature is to replicate these latent grouping strategies for subgroups of students (e.g., first-generation learners or racial or ethnic minority groups) to build a more comprehensive predictive identification of the level of test anxiety that should be targeted as the “cut score” for providing outreach and intervention services to learners with different backgrounds or experiences.

Finally, while the availability of clustering and latent class procedures shows great promise for identifying levels of test anxiety, these are not necessarily the optimal approaches. As mentioned, there were no identified studies in our review that used individual item characteristics to determine cut scores for levels of anxiety for post-secondary learners. However, prior research with younger populations shows that these strategies (e.g., ROC and RMS; see [63]) have a great advantage in ensuring that the most relevant items or subscales in a measurement tool contribute the most to the final determination of the criterion for identifying learners’ levels of test anxiety.

A final note on this recommendation is to highlight that we recommend “Adopt and Adapt”; that is, merely adopting existing values from prior studies is not necessarily the optimal solution for building a diagnostic procedure in an academic context where the goal of the institution is to identify learners with the greatest needs for intervention related to test anxiety supports. As the student body differs from the original study location and the new implementation location, adjustments may be required. As such, employing a validation check periodically is critical. To reiterate, while the TAI was the most commonly used measure for leveling studies in our review, those that employed the published criteria for “high test anxiety” were based on the 1980 norms, which are certainly out of touch with the contemporary post-secondary student population.

### 4.3. Limitations and Future Directions

This systematic review is limited in the ability to reliably offer a “solution” for defining “clinical” test anxiety or setting the standard for determining what level or degree of test anxiety should be considered detrimental for learners. However, the primary purpose of this study was to establish the “field as it is”, which hopefully will enable the community of scholars focused on this work to take on the next step—identifying a common definition for elevated or clinical levels of test anxiety. The primary limitations we observed in this process are the diversity of assessment measures, the variations in leveling strategies used in prior work, and the wide variations in student populations addressed in this work. Our observation across the post-secondary population in this work has revealed an unexpected lag in the research with this population. That is, there is a far lower degree of attention to establishing “levels” of test anxiety for university students to identify high test anxiety for support and treatment than for younger populations (i.e., children and adolescents; see [57]). This is an area of growth for the field that appears in line with the current needs in higher education retention goals. As more students are attending post-secondary training, the number of individuals who are likely to feel stressed or overwhelmed by test anxiety is increasing. If institutions of higher education intend to support these students (if for no other reason than to maintain enrollment levels), employing diagnostic assessments to identify learners with heightened needs is one avenue for institutional priorities as well as a vibrant area for program efficacy research (see [8], for an extended discussion).

## Figures and Tables

**Figure 1 behavsci-15-00331-f001:**
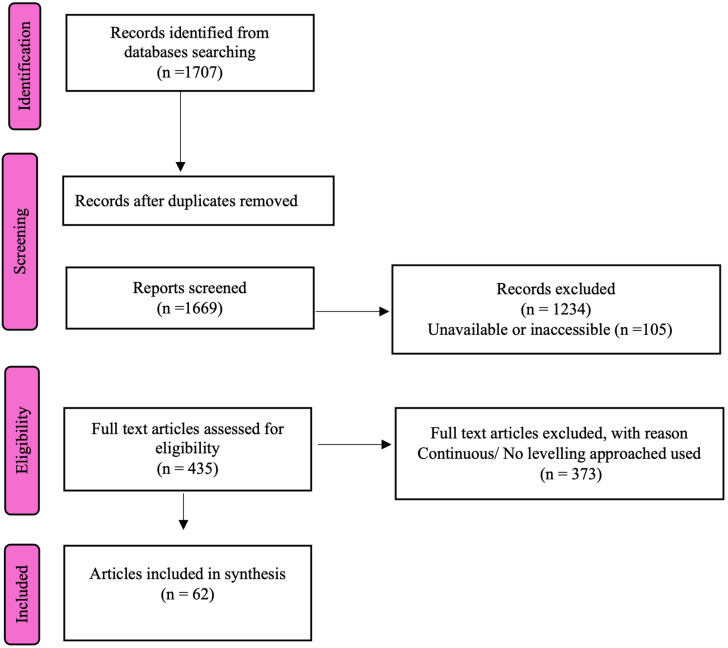
PRISMA table for review.

**Table 1 behavsci-15-00331-t001:** Test anxiety measures and methods of leveling.

Test Anxiety Instrument	Number of Studies	Split	Val	LogCut	Cluster
China Language Anxiety Survey (CLAS)	1	-	-	🗸	-
Cognitive Test Anxiety Scale (CTAS)	5	🗸	🗸	-	🗸
Foreign Language Classroom Anxiety Scale (FLCAS)	6	🗸	-	🗸	-
German Test Anxiety Inventory (PAF and PAF-E)	4	-	🗸	-	-
Hamilton Anxiety Scale	1	-	🗸	-	-
Medical Student Stressor Questionnaire (MSSQ)	1	-	-	🗸	-
Motivated Strategies for Learning Questionnaire (MSLQ)	1	-	-	-	🗸
One-Off Scales Created by the Researcher for the Current Study	4	🗸	-	🗸	🗸
Sinha Anxiety scale	1	-	-	🗸	-
State-Trait Anxiety Inventory (STAI)	4	-	🗸	🗸	-
Test Anxiety Inventory (TAI)	14	-	🗸	🗸	🗸
Test Anxiety Questionnaire (TAQ)	3	🗸	-	🗸	-
Test Anxiety Scale (TAS)	6	-	-	🗸	🗸
The Express Test–The Diagnostics of Examination Anxiety	1	-	-	🗸	-
Visual Analog Scale (VAS)	1	🗸	-	-	-
Westside Test Anxiety Scale (WTAS)	8	🗸	🗸	🗸	-
Zung Self-Rating Anxiety Scale	1	-	-	🗸	-

Notes: Identified leveling methods: Cont—test anxiety measured as continuous variable, no leveling. LogCut—logical cut scores, not data-driven. Val—validated, using standardized methods or published leveling values. Splits—data-based splits using “local” samples’ median, mean, etc. Cluster—student-focused grouping methods.

## Data Availability

Data are contained within the article and Appendix A.

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
