# Peer review of "Methods Employed in Studies Identifying “Levels” of Test Anxiety in University Students: A Systematic Review"

_behavsci, 2025, doi:10.3390/bs15030331_

Round 1
Reviewer 1 Report
Comments and Suggestions for Authors
1. How do you explain the selection only of those databases? Please explain.
2. A flow chart indicating the process followed would help a lot understant the study.
3. "Attempting to offer some basic guidelines for building a set of standards" this is ambiguous. The study does not deal with standards at all.
4. "The results of our review confirmed that the vast majority of 21 studies employ strategies that are not optimal" for this statement you should have some hypotheses probably. The study does not have.
5. What are the implications of the study?
6. Citations are in APA style now. Check the citation standards.
Author Response
Thank you for the careful review of our work. The review process was an excellent exercise in helping us identify where we had failed to communicate clearly as well as identify areas of boosting the utility of the paper.
Below are point-by-point responses to comments from the reviewers. We hope that the explanation of these decisions and adjustments align with the requests you suggested. If we have missed the mark, please advise. You will see commensurate revisions throughout the article, all marked as tracked changes in the document (unless noted below for simplified formatting with the tables and figures in particular).
Reviewer 1:
- How do you explain the selection only of those databases? Please explain.
The decision for these databases was guided by two primary factors. First, these databases cover the domains of inquiry primarily encountering the literature on test anxiety research – and aligns with a similar systematic review that we are replicating (Tan et al., 2025). Second, our support from the library experts at our institution confirmed that the data captured in this approach would be a comprehensive method for the parameters we established (peer review, published, English language, etc). This is addressed in the methods section now.
- A flow chart indicating the process followed would help a lot understant the study.
I apologize – the PRISMA chart appears to have been dramatically damaged in the translation to the version the reviewers received. We have changed the method of showing the PRISMA (Figure 1) in the hope that this corrects the problem. Note – to simplify the viewing, we did not “track” this change or mark it in a different color.
- "Attempting to offer some basic guidelines for building a set of standards" this is ambiguous. The study does not deal with standards at all.
Thank you for identifying a poorly constructed statement. We have adjusted the language around that sentence in the abstract. Indeed, our focus is to establish suggestions of what researchers “should do” and “should not do” (or what readers should or should not accept as valid) when attempting to identify different “levels” of test anxiety in their populations.
- "The results of our review confirmed that the vast majority of 21 studies employ strategies that are not optimal" for this statement you should have some hypotheses probably. The study does not have.
There indeed were no specific hypotheses, which we believe is appropriate for this approach. However, the point you make about the conclusion offered here raised our attention to see that indeed we should reframe that statement as more descriptive.
- What are the implications of the study?
Thank you for raising the question – we have rewritten the discussion section to address this. In short, we believe: 1. There are too many variations in approach to determining “high anxiety”; 2. The strategies used in much of this work do not follow preferred methodological strategies; 3. It is relatively easy now to do this work and build a commonality.
- Citations are in APA style now. Check the citation standards.
All citations have been updated to be consistent with APA – and DOI information included when available. These changes are not marked with “track changes” due to the messiness that ensues with all these revisions.
Reviewer 2 Report
Comments and Suggestions for Authors
The topic is interesting, and it is always desirable to have a methods review; yet it should take more care to give it a title that refers to the work done: it is only a methods categorization. The relevance part should be based closer to the literature. The study selection should be described in more detail. The samples and methods used in the reviewed studies should be described. There should be a real path or at least ideas how to solve the presented problems.

Author Response
Thank you for the careful review of our work. The review process was an excellent exercise in helping us identify where we had failed to communicate clearly as well as identify areas of boosting the utility of the paper.
Below are point-by-point responses to comments from the reviewers. We hope that the explanation of these decisions and adjustments align with the requests you suggested. If we have missed the mark, please advise. You will see commensurate revisions throughout the article, all marked as tracked changes in the document (unless noted below for simplified formatting with the tables and figures in particular).
Reviewer 2:
The topic is interesting, and it is always desirable to have a methods review; yet it should take more care to give it a title that refers to the work done: it is only a methods categorization. The relevance part should be based closer to the literature. The study selection should be described in more detail. The samples and methods used in the reviewed studies should be described. There should be a real path or at least ideas how to solve the presented problems.
Several key points offered in this opening statement. We appreciate the overviewing guidance, and hope to identify specific points in response below.
- We have adjusted the title to be more precise on our focus on the methodological variations in the field – that is indeed our goal.
- Study selection detail has been improved in three ways. First, we have put in a more visiable PRISMA table (to show the process). Second, we have provided more elaboration around our search and selection criteria. Third, we point to the “supplemntal” materials more directly in the methods, results, and discussion – calling attention to the global nature of the work, the large variations in samples, etc.
And…
General Aspects
1) The title and abstract raise expectations that the text does not fulfil: ‘Systematic review’ to identify anxiety levels in test anxiety sounds like a systematic review of anxiety studies in university students. This is also formulated in the abstract as follows: ‘examining the state of research in the field’ and ‘offer some basic guidelines for building a set of standards’.
Thank you – your guidance toward being more direct in stating plainly what our study and our goal is focused on is indeed a critical point we have taken pains to follow. Indeed, revision of the title and abstract was critical (and led to commensurate adjustment to the rest of the paper).
2) However, this is by no means the case: it is a purely methodological observation under certain, set assumptions:
- It would be good to have a clinical sample on which tests can be validated. Because of the ambiguous definition and because test anxiety does not occur as a single diagnosis in the DSM, such validation is not possible or cannot be found.
- Identifying the levels of test anxiety is important because it makes it possible to identify people with high test anxiety so that they can be offered help. As their test anxiety decreases, they become more successful.
Both assumptions are problematic: on the basis of the first, all studies that have continuously conceptualised test anxiety are excluded due to methodological considerations; yet it is precisely the results obtained in this way that show possible influencing factors.
We agree with the importance of examining test anxiety as a continuous measure when building a working understanding the “causes and consequences” of test anxiety. Most of our work indeed uses this orientation – and our primary focus with test anxiety research lately has been focused on curvilinear relationships in particular. However, those relationships lie outside the focus of this more “clinical” approach to discussing test anxiety – and required the removal of those studies (as they did not employ any leveling methods).
To your point, however, we have added statements in the literature review clarifying this disconnect between the “applied leveling” and “continuous” application of the data in test anxiety. We have also attempted to make more plain this disconnect between “applied” and “theoretical” approaches to the literature (which we observe now is MORE prevalent in the post-secondary population.
The second is even more problematic because it often involves at least effects in both directions: test anxiety is shown to be a consequence and not a cause (see below). – we address each point below.
The studies used refer not only to university students, but also to college students.
Thank you – this was a cultural language oversite. In the US (lead author’s location), we often use the terms interchangeably to refer to any students in a 4-year post-secondary institution (the difference is technically an arcane one related to graduate study historically). We have adjusted this mistake to be more globally relevant.
In a ‘systematic review’, I would expect an overview of theoretical approaches to test anxiety.
We hope the adjusted promise of the title and abstract point toward the intent of the review more accurately here. While a brief overview of models of test anxiety apply (and we believe is offered), a full accounting of this work has been addressed elsewhere and falls outside what we believe is the scope. Guided by some of your other theoretical notes below, we have expanded the view of theory.
However, this is missing or at least very incomplete:
Theoretical Part
- The correlation found between test anxiety and test performance is ‘present, but generally very low; the direction of effect is unclear: it has been found that a lack of self-efficacy and test anxiety are also fuelled by the experience of failure, for example due to social or cognitive deficits (such as knowledge gaps) (e.g. Verissimo 2022).
We do not fully agree with your interpretation of Verissimo et al’s analysis. For the point “direction of effect is unclear” – agreed. Indeed, comments from that paper identify that “results confirm that AA is a predictor of anxiety” and “confirm an association between low AA and Anxiety” and that the low AA is a “risk factor for higher anxiety” (etc). However, their study has the same limiter that they argue about – that is, they only tested the direction one way (not both). I agree they have a reasonable point that is critical in that directionality of the association is not one-way. They also report that “test anxiety leads to poor academic performance” as well in their paper. In essence, Verissimo offers the view that there is a bidirectional relationship – which I agree with (I offer a more “circularity” notion, that there is a recursive circle in this). The point you offer that “generally very low”).
Regarding the “relationships is present but generally very low” – it really depends on how one is defining the relationship and “low.” Indeed, this sounds more like the argument that Jerrim offers repeatedly – that there is merely a “weak” relationship between test anxiety and performance. Using Cohen’s guidelines, that is correct. However, it is still explaining between 10 and 20% of the variance in performance in most studies. While that is not “moderate” or “large” – it’s meaningful (we argue) – particularly when you are using a variable like anxiety to account for a measure of ability and achievement that is influenced by the range of human influences.
Regardless of our theoretical difference of opinion on this – we agree that more attention was warranted to these points – and have addressed this in the literature review. To that point, we have identified the reality of imprecise direction of relationship as well as the need to know as many things about the learners as possible (in the literature review) when building a full understanding of human performance as related to academic anxiety. Interestingly, the Verissimo study uses a leveling procedure themselves – but they level academic achievement (not anxiety)
- The statement of an increase in the proportion of students with a self-report of exam anxiety from 30% to 80% in different studies and measurement methods seems dubious to me, especially in a text that otherwise focuses strongly on methods.
Indeed, this is the usual response when I share these data in professional settings. We have provided more detail on this rather large number and diverse range of findings. Frankly, it’s surprising to my research team as well…but is the foundation of our current studies, which is demonstrating that there is a significant increase in the level of reported test anxiety among university (post-secondary) students. I can’t cite the following, but my conversations with leading exam anxiety scholars around the globe have all confirmed they are seeing similar increases. As for the “spread” – this is explained in part by “when” the data were collected (older data have lower percentages of “test anxious” students) – as well as the “method” – a key part of this study goal, to help identify that these variations need to be addressed. We hope our treatment of this in the text has clarified the purpose of pointing this out, as well as captured your primary concern about a seemingly outlandish value of self-reported test anxiety.
It is stated that most studies work with self-report questionnaires that are based on a theoretical model and that they methodologically assume continuous measurements and normal distribution and that training studies recognise the success of training in a change in the corresponding scales. As evidence of relevance for their diagnostic approach, it is stated that the interventions for test anxiety in university students led to greater well-being and better performance, but: Both authors cited have not studied university students: In v.d.Embse's study 10th graders participated, in Putwain's college students.
Thank you for the point. We have clarified and moderated that claim. Unfortunately, very few systematic studies are available for the post-secondary group, which is a point of specific mention in our discussion section now.
The categorisation between emotionality (or physiological reactions) and cognitive components is too simple. ‘In the end, when practitioners are attempting to identify learners with heightened levels of test anxiety in need of support, the most common focus is on the ‘composite’ or ‘total’ representation of test anxiety that incorporates both of these broad dimensions (Friedman & Bendas-Jacob, 2008; Lowe et al., 2011; Sarason, 1994; von der Embse et al., 2013).’ This statement is incorrect: there is a consensus that the physiological component correlates least with poor performance. (e.g. Cassidy & Johnson, 2002). The authors mentioned in the text do not find two components: Friedman & Bendas-Jacob (2008) describe three components (physiological, cognitive and social); Lowe et al, 2011 find four factors (physiological, social, task-irrelevant behaviour, rumination), and they do not identify learners in need of help, but investigate the construct of test anxiety.
The preference for a bifactor model is indeed a historical preference – not universally agreed upon model in today’s research context. We have attempted to offer a less definitive statement and recognize that the field is diverse in this domain. However, a review of these articles will identify that the vast majority of people doing this work are indeed still maintaining the Spielberger or Liebert/Morris orientation. To your point – the bulk of the work you identify is not focused on “leveling” students, and attempting to identify those who are at risk or in need of supports/interventions. That is the purpose of this discussion – to call attention to the clinical desire to have this information.
More recent approaches are based on four test anxiety components (emotionality, worry, interference and lack of confidence), which are linked to performance in different proportions.
I would differ with saying that “more recent” approaches do this. This approach was dominant in the late 1980s and early 1990s, and has come back into vogue in a few sectors (including Putwain and VdE as well as FriedBen and others – but their work is not with post-secondary as mentioned). Other attempts to revitalize Benson, Sarason, etc have actually demonstrated in their factor analyses that at least “cognitive interference” does not fit the model – and differentiating among worry, tension, and lack of confidence are not readily reconciled.
Again – though, we take your point – and have adjusted the section to be less dogmatic in our offering. The overview of assessments now identify the wide variety of approaches, and do not assert a dominance of the bifactorial model.
Motivation also plays an important role here, as can be seen, for example, in different performance goals (see, for example, Moecklinghoff et al. 2023). However, even older studies (e.g. Rost & Schermer 1992) come up with more components (here 5: Worry, tension, irrelevant thinking, physiological reactions, emotionality).
We have added in the control-value theory in particular here, which is a strong representation of this work in the literature review.
The overview of the measurement methods is systematic, but already mixed with criticism that is not really resolved. Some of the criticism is actually unspecific and applicable to almost all self-report questionnaires. However, there is actually no real alternative. The prioritised approach (ROC) also requires a clear clinical classification and Rasch scaling requires one or more dimensions in which items can be clearly classified. According to the authors, both are missing for the construct of test anxiety.
You are correct, that is our claim. The ROC work has been done – but not with the population in question. And, there is no clinical definition (so we can’t do that work).
Methods
The selection of articles found via databases is not clear enough. Only studies in which test anxiety was actually measured were selected. In this way, there were n=435 studies. Of these, 373 were excluded because they did not measure any leveling of the construct. The 62 remaining studies were then categorised again into four (to five) categories with relatively high interrater agreement (1) local sample splits, (2) person-focused data-based grouping (clustering) (3) published or validated splits, (4) scale based cut scores validated by experts. After presenting the categories it is described how many of the 62 selected studies fall into which categories and examples are given.
WE have added to the methods section on this – and clarified the PRISMA table as noted before. We appreciate the suggestions (and you have largely identified the crux of our findings in concurrence with our intention here).
It is considered why ROC and Rasch-scaled measurement methods are not found in the 62 selected studies. It is stated that ROC was found in von der Embse et al. 2018; I consider this to be a false assertion, as I did not find any ROC scaling in this meta-analysis(!) when I looked it up.
This citation to VdE was inaccurate (by our team). The correct citation is VdE 2021, and has been adjusted. We are appreciative that your careful review caught our embarrassing error.
What follows is the presentation of Table 1, which refers to the larger set of empirical studies on test anxiety (n=435). This table is completely incomprehensible to me and seems contradictory to the rest of the text. It raises many questions for me: Are there 75 studies on the TAI in which the three evaluation methods mentioned were used? Are there 75 studies on the TAI that also included studies that used the aforementioned evaluation methods? How many were there in each case? Why are the samples not mentioned? Why does the number of studies only add up to around 400 and not to 435?
We have recreated Table 1. It was terribly confusing and not helpful as mentioned by the reviewer. To the point about the numbers – and your next point – there were studies that used multiple TA measures.
Are there no studies that used more than one test instrument?
There are. But – those studies use one instrument to “level” and the others to validate. We have added this explanation to clarify.
The table is so incomprehensible is perhaps due to the fact that the explanations in the text are very sparse.
In addition to changing the table, we have bolstered the explanation of what the table conveys in the text (as well as point to the supplementary materials, where more details are available for each study.
Discussion – we have broken up the comments here to direct our responses:
The discussion is also very brief.
Thank you for your point – this is a common issue in our team (we tend to “underexplain” in the discussion. More context has been added, we hope without over-reaching.
The aim of the article to ‘provide methodological guidance to support more appropriate and valid strategies’ was not fulfilled, because no good solutions to the problems mentioned are presented.
Indeed – this was the primary thrust of additions. We have restructured the discussion to explicitly identify our recommendations directly. Thank you for calling attention to this unfulfilled promise of the paper.
It is not even well documented why a sample-independent leveling approach to diagnosing test anxiety could help with students' performance problems. The figures are also repeatedly misleading: only 3.7% of studies are said to use leveling scaling, but this is based on a population of studies of which around 75% are apparently not empirical anyway. If only the empirical studies form the population, the figure is still 14%. Incidentally, only those who underestimate the theoretical models can underestimate these 75% of studies that ‘only examined the relationships with other relevant educational or psychological variables’; it is less a question of identifying those with high test anxiety via a ‘level’ criterion that applies to all, but rather of better understanding the causal structure around test anxiety, self-efficacy, procrastination, motivation and (meta-)knowledge in relation to test performance on a population-specific basis. Each of these parameters can be addressed in order to improve performance. Test anxiety does not necessarily stand out here.
Excellent observation on the appropriate number for comparison – we have adjusted.
We have also commented on the fact that test anxiety is merely ONE of the many affective variables of possible interest that may influence the performance or experience of students in higher education. Furthermore, we have expanded the discussion of how we interpret the potential of identification and intervention strategies in higher education may promote student outcomes, and why the appropriate selection of a “leveling criterion” influences this process overall.
Again – we appreciate the very detailed and thoughtful analysis of our gaps in this reporting, and believe that your suggestions have helped us better frame and clarify our message.
Conclusion:
In addition to the overall problem that the relevance of the specific question for the specific population (university students) was not well elaborated and the self-imposed requirements are not met, the approach is often not clear, many references and assertions are incorrect or inaccurate. The text appears unfinished or carelessly edited.
A minor point to note is that the format of the references is also inconsistent.
All references have been updated to be more accurate and APA compliance.
A full review of the article for clarity has also been undertaken – and we agree it was needed.
Round 2
Reviewer 1 Report
Comments and Suggestions for Authors
Thank you for all the changes.
Author Response
Reviewer Comments: Thank you for all the changes.
Response - many thanks for a careful and engaged review process!